# Laboratory and Metabolomic Fingerprint in Heart Failure with Preserved Ejection Fraction: From Clinical Classification to Biomarker Signature

**DOI:** 10.3390/biom13010173

**Published:** 2023-01-13

**Authors:** Alberto Palazzuoli, Francesco Tramonte, Matteo Beltrami

**Affiliations:** 1Cardiovascular Diseases Unit, Cardio Thoracic and Vascular Department, Le Scotte Hospital, University of Siena, 53100 Siena, Italy; 2San Giovanni di Dio Hospital, 50143 Florence, Italy

**Keywords:** biomarkers, metabolomic, microRNA, HFpEF

## Abstract

Heart failure with preserved ejection fraction (HFpEF) remains a poorly characterized syndrome with many unknown aspects related to different patient profiles, various associated risk factors and a wide range of aetiologies. It comprises several pathophysiological pathways, such as endothelial dysfunction, myocardial fibrosis, extracellular matrix deposition and intense inflammatory system activation. Until now, HFpEF has only been described with regard to clinical features and its most commonly associated risk factors, disregarding all biological mechanisms responsible for cardiovascular deteriorations. Recently, innovations in laboratory and metabolomic findings have shown that HFpEF appears to be strictly related to specific cells and molecular mechanisms’ dysregulation. Indeed, some biomarkers are efficient in early identification of these processes, adding new insights into diagnosis and risk stratification. Moreover, recent advances in intermediate metabolites provide relevant information on intrinsic cellular and energetic substrate alterations. Therefore, a systematic combination of clinical imaging and laboratory findings may lead to a ‘precision medicine’ approach providing prognostic and therapeutic advantages. The current review reports traditional and emerging biomarkers in HFpEF and it purposes a new diagnostic approach based on integrative information achieved from risk factor burden, hemodynamic dysfunction and biomarkers’ signature partnership.

## 1. Introduction

Heart failure with preserved ejection fraction (HFpEF) is a heterogenous syndrome with specific molecular, genetic and metabolomic features, all of which reflect on vascular and myocardial cell adaptations [1]. HFpEF encompasses different pathophysiological pathways and cardiac structural profiles compared to heart failure with reduced ejection fraction (HFrEF) [2]. About half of individuals with heart failure (HF) are considered to be affected by HFpEF showing a peculiar clinical profile and cardiac structural and functional alterations. Therefore, the selection criteria are often elusive and mainly based upon ejection fraction cut-offs rather than distinct clinical and laboratory phenotypes [3]. Most inclusion criteria comprise the concomitant presence of left ventricular hypertrophy (LVH), altered diastolic dysfunction and elevation of serum natriuretic peptide (NP) levels associated with exertional dyspnea or reduced exercise tolerance. Indeed, recent clinical trials have adopted wide inclusion criteria and patient features creating inhomogeneous patterns with various morphologies and comorbidities [4,5]. In this framework, advanced analytic research, investigating specific biomarkers in a well-phenotyped population, could lead to better understanding about molecular pathways and biological mechanisms responsible for HFpEF syndrome. The interaction between clinical variables, imaging features and biomarkers could become the model for future research and a combined network analysis may change the current approach based on traditional knockdown/knockout study [6]. In HFrEF syndrome, myocyte loss, cellular death and consequent cardiac chamber enlargement are the main features causing the disease progression; conversely, HFpEF is characterized by collagen overexpression, myocardial fibrosis, extracellular matrix deposition and high inflammatory response [7]. All of these mechanisms occur differently according to specific risk factors, comorbidities and vascular and cardiac remodeling [8]. Thus, an analysis based on detailed phenotyping, a cardiac structural alteration and distinctive laboratory findings may challenge the current scenario, leading towards a precision medicine model with specific therapeutic targets considering different individual profiles [9]. This personalized setting begins from machine learning analysis of big data in order to resolve disease heterogeneity by identifying patients within particular subtypes and predicting response to the therapy.

## 2. Different HFpEF Phenogroups

Despite recent improvements in treatment and diagnosis, HFpEF remains a poorly characterized syndrome with many unknown aspects related to different patient profiles, associated risk factors and pathophysiological pathways [10]. Vast trials have shown a wide prevalence of LVH left atrial dilatation, diastolic dysfunction and post-capillary pulmonary hypertension. Pulmonary hypertension (PH) is common in both patients with HFpEF and HFrEF and is associated with higher hospitalizations and mortality. Observational studies suggest an estimated prevalence of PH of 40–72% in patients with HFrEF and 36–83% in those with HFpEF [11]. Moreover, HFpEF patients have presented various extracardiac comorbidities such as diabetes, chronic kidney disease (CKD), anemia, chronic lung diseases, obesity and metabolic dysfunction [5,12,13]. Because the interventional trials did not distinguish between different risk factors and underlying diseases, the one-size-fits-all approach might explain the lack of efficacy and benefit of current treatments. Based on the different pathophysiological drivers, some authors have suggested that different HFpEF subtypes are linked to cardiometabolic alterations, body structural conformation and peripheral maladaptation. These assessments may be related to the presence of systemic disorders leading to skeletal muscle metabolism alterations and vascular rarefaction [14]. Since all of these features are widely expressed in HFpEF, the diagnosis based only on cardiac morphology and dysfunction remains difficult to interpret and is often misleading. Current pictures may configure a wide range of HFpEF phenotypes which differ in cardiac structure and cardiovascular remodeling, both related to the underlying biological process and pathophysiological contributor, despite having a similar EF.

Notably, recent machine learning analysis has attempted to cluster specific phenotypes by latent class study. In a post hoc analysis of TOPCAT, patients were classified into three categories according to vascular and cardiac remodeling: patients with mild LV hypertrophy and chronic pulmonary disease with normal vascular stiffness characterized by increased expression of metalloproteinase; older patients with multiple comorbidities, LV hypertrophy and reduced vascular compliance, characterized by an elevated tissue calcification biomarker; and obese subgroup with several metabolic alterations, increased renin–angiotensin system activity, lipidic profile derangement and increased inflammatory pattern [15]. Similarly, another study identified a group including young individuals with increased body mass index (BMI), typical abnormalities in cardiac structure and function and low serum levels of natriuretic peptides (NP); a cluster with high prevalence of diabetes and obesity characterized by severe diastolic dysfunction and elevated NP circulating levels; and a cluster characterized by RV dysfunction and combined pre- and post-capillary pulmonary hypertension and renal dysfunction, experiencing the worst outcomes [16].

Finally, an analysis of SwedeHF and CHECK HF registries differentiated between five distinct phenotypes according to a combined approach including risk factors and associated comorbidities. The study confirmed that the cluster with CKD, coronary artery disease (CAD) and high diuretic amount revealed the worst outcomes [17] (Figure 1).

These data reveal some common features but also some discrepancies underlying the need for a more homogenous classification. Moreover, a detailed screening capable of identifying a specific cluster through the combination of structural cardiac abnormalities and the underlying molecular mechanisms responsible for the progression of this disease may lead to a better understanding of this syndrome and therefore to specific therapeutic target opportunities [18].

## 3. Current Biomarkers in HFpEF

Several HF risk prediction scores include biomarkers, mainly natriuretic peptides (NP), but important gaps exist regarding the knowledge of the underlying pathophysiological mechanisms, biological process and disease progression. Circulating biomarkers should reflect cardiac and extra cardiac disorders responsible for the HFpEF development and the related pathological pathways [19,20]. Since HFpEF is characterized by LVH and increased parietal stress, systemic vascular damage and stiffness, increased inflammation and enhanced fibrosis, we recognize four main biomarker targets: myocardial injury, extracellular fibrosis, inflammation and endothelial dysfunction (Table 1).

**Myocardial Injury**—High sensitivity troponin (HsTn) is universally considered a marker of myocardial damage in acute coronary syndrome (ACS). However, it has prognostic significance in HF and it implies myocardial damage apoptosis and progressive fiber loss independent of coronary vessel diseases. It could be the final outcome of microvascular dysfunction and subendocardial layer damage due to systemic oxygen reduction and therefore an altered supply–demand mismatch. Other features, such as increased wall tension, high left ventricle (LV) filling pressure and right ventricular dysfunction, are related with increased HsTn levels [21]. In patients with HFpEF, increased HsTn serum levels correlate with a more severe diastolic degree and a higher pulmonary pressure. Moreover, high HsTn serum levels are also associated with an increase in wall stress, a higher degree of LV hypertrophy and an increase in cardiac workload [22]. Many reports have shown that HsTn predicts poor outcomes in HFpEF, especially in men rather than women. In hospitalized patients with acute HfpEF, the persistence of high HsTn serum levels at both admission and discharge is related with increased rates of rehospitalization and death [23,24]. Similarly, in the TOPCAT trial, elevation of HsTn was independently associated with a higher risk of hospitalization and cardiovascular events [25]. These findings were confirmed in the PARAGON study in which even a mild HsTn elevation was associated with a worse outcome during a follow-up of about three years; moreover, patients taking sacubitril/valsartan treatment showed a significant reduction compared to the placebo [26].

Natriuretic peptides (NPs) are the hallmark biomarkers in HF and their measurement is accounted for in HF guidelines across the spectrum of the whole EF [27]. The biologically active NP form and its amino-terminal portion precursor (pro B type natriuretic peptide) are cleaved into NT-proBNP and BNP and released in response to enhanced cardiac wall tension and increased filling pressure; moreover, their levels increase proportionally to the degree of systolic dysfunction. The two peptides are released in response to sympathetic activity, in particular to systemic vasoconstriction and fluid retention, as an opposite response to the increased neurohormonal overdrive [28]. NP activity counteracts sympathetic activity promoting cardiac afterload reduction and myocardial relaxation by directly eliciting vasodilatation and myocardial relaxation effects. The main mechanism of action is related to diuresis and natriuresis that lead to congestion reduction and euvolemia [29]. Serum levels of NPs are directly related to intracardiac pressure, including LV end diastolic pressure (LVEDP), wedge pressure and pulmonary systolic pressure. Both peptides are largely analyzed in patients with reduced systolic function as valuable diagnostic and prognostic features. In HFpEF serum, NP levels are generally less increased but they keep their diagnostic relevance [30]. Some authors believe that this feature is due to the reduced wall stress in this setting together with extracardiac conditions such as metabolic syndrome, chronic lung disease and, in particular, obesity in which adipocyte cells favor a reduced NP receptor expression [31]. These comorbidities are often associated with one another in HFpEF causing a wide range of NP levels. Although some studies have revealed that some HFpEF clusters experience low NPs below 100 pg/mL, a recent meta-analysis has shown an optimal diagnostic accuracy in this setting (AUC 0.80 CI 0.73-0.87) [32]. Moreover, a combined analysis of NPs and HsTn has shown that patients with higher serum levels have an increased risk of death and hospitalization [33]. Finally, in acute settings, NP assays reveal similar prognostic information in HFpEF as in HFrEF and the related changes during hospitalization confer equal risk assessment adjusted for potential confounding factors [34].

Adrenomedullin (ADM) is a regulatory peptide produced by endothelial and smooth muscle cells with antiproliferative, vasodilatatory and antiapoptotic effects. It is synthesized mainly by adrenal medulla but its receptors are expressed in many tissues such as lungs, heart and kidneys [35]. It is considered an important biomarker of pulmonary and systemic congestion and it is produced in relation to increased sympathetic activity [36]. It counteracts systemic vasoconstriction induced by renin angiotensin system activation facilitating vascular permeability and elastance. Due to its serum instability, mainly caused by interactions with plasma proteins, and short half-life, a reliable quantification of ADM is difficult to achieve and its precursor ‘mid regional pro-hormone’ (MRpro-ADM) is usually measured [37]. A large study confirmed the close relationship between ADM and congestion in patients with worsening heart failure; therefore, a high plasma level appears to be related to increased risk and recurrent hospitalization for HF [38]. MRpro-ADM measured at admission is also related to all causes of cardiovascular mortality, sudden death and cardiac arrest [39]. In patients with acute coronary syndrome (ACS), elevated MRpro-ADM levels predict the risk of HF occurrence. Finally, in the PROTECT trial, the MRpro-ADM was related to longer hospitalization, increased congestion signs and elevated NP levels; moreover, its assessment before discharge conferred relevant prognostic information related to incomplete decongestion status and therefore early rehospitalization risk [40].

**Extracellular Fibrosis**—Collagen deposition and increased myocardial fibrosis are two relevant features in HFpEF. The more extensively analyzed biomarkers of this process are galectin-3 and soluble ST2. Galectin-3 is a glycoprotein involved in many inflammatory and profibrotic processes as a galactosidase family member, and it is synthetized by macrophage [41]. It directly increases fibroblast proliferation and fibrogenesis in animal models, inducing myocardial and vascular stiffness. It is also associated with renal dysfunction and LV remodeling [42]. Galectin-3 inhibition mitigates myocardial fibrosis and elicits a reverse remodeling through a reduction in systemic overload [43,44]. High galectin-3 serum levels are associated with poor outcome in both patients with HFrEF and HFpEF. In patients with elevated levels, galectin-3 is associated with other comorbidities, such as hypertension and CKD, and it is a useful marker for target therapy and risk stratification [45]. Moreover, changes in galectin-3 levels over a period of follow-up provide prognostic insights in patients with HFpEF [46].

Soluble ST2 is another marker reflecting myocardial fibrosis and it is overexpressed in HFrEF and HFpEF patients. It is primarily produced by myocardial cells, but also smooth muscle and endothelium cells, in relation to congestion or profibrotic stimuli [47]. In HFpEF patients, the addition of ST2 to NPs provides more complete prognostic information; therefore, a higher ST2 phenotype could indicate a more compromised diastolic dysfunction [48,49]. Notably, a meta-analysis demonstrated that ST2 could predict outcomes independently of EF values [50].

Matrix metalloproteinases (MMP) and tissue inhibitors of metalloproteinases (TIMP) are two endopeptidases which induce extracellular collagen deposition; therefore, they are reasonably considered as two biomarkers of fibrosis in HFpEF [51]. Collagenase is an enzyme family with different characteristics and may be considered in the context between collagen synthesis and collagen degradation. Elevated levels of MMP2 and MMP9 are related to an increased risk in HFpEF but also high levels are found in HFrEF after myocardial infarction [52]. In the PARAGON trial, a high level of TIMP, a marker of impaired collagen degradation, is associated with increased event rate [53].

Additional collagen biomarkers such as procollagen type I (PIP) and procollagen type III N-terminal peptide (PIIINP) demonstrated a predictive role in high risk patients for HFpEF development. Both biomarkers reflect increased collagen deposition and turnover. They appear to be associated with the extent of collagen deposition in myocardial biopsies [54]. However, cross-sectional analysis showed contrasting results: in the Framingham sub-study, PIIINP was not associated with echocardiographic abnormalities, whereas in the Cardiovascular Health Study it was associated with an increased risk of incident HF [55,56].

**Inflammation**—Systemic inflammation is a typical feature of HFpEF. It reflects the immune response to cardiac remodeling, systemic vascular injury and underlying triggers often associated with diseases, such as metabolic syndrome, diabetes, chronic lung disease and anemia [57]. Inflammation can occur differently in every HFpEF phenotype and it can be analyzed using several biomarkers. C-reactive protein (CRP) is the wider analyzed marker and it is associated with an increased risk in ACS and HF. A comparison study differentiating CRP from HFrEF and HFpEF has demonstrated that in the latter it has a better prognostic meaning, adding new information rather than just that of NPs alone [58]. CRP and pentraxin are significantly higher in acute HFpEF patients compared to non-acute patients and they correlate with diastolic dysfunction degree [59]. CRP has a direct role in inducing complement cascade activation and cytokine stimulation causing myocyte loss and endothelial dysfunction by decreasing nitric oxide (NO) production. A CRP increase is also related to immune response mediated by lymphocyte T and monocyte cells. Inflammatory status may also trigger microvascular dysfunction by inducing endothelial permeability and adhesion molecule production and increasing reactive oxygen species bioavailability [60]. However, elevated CRP levels were observed in acute and chronic diseases, and both infectious and non-infectious diseases, indicating an acute or persistent inflammatory response. Mean levels varied according to the disease and indicated a baseline level in the individuals with a particular disorder. The clinical significance of CRP should be counterbalanced in the clinical context by evaluating the presence of infections and/or chronic inflammatory diseases in HFpEF patients.

Grow differentiation factor 15 (GDF-15) is a member of the cytokines family and it belongs to the transforming growth factor beta (TGFβ) family. It is highly expressed in inflammatory chronic diseases, and pulmonary, kidney, and cardiovascular diseases [61]. Since it integrates information from cardiac and systemic diseases, it could reflect the interplay among different apparatuses, but it is not specific to CV diseases or HF [62]. A recent meta-analysis demonstrated that in patients with a high risk burden it is related to an increased incidence of HF providing additional information on LV remodeling and function [63]. In HFpEF, it is similarly elevated as in HFrEF but it has a more prognostic value compared to NTproBNP. Indeed, in subjects with low NT-proBNP and high GDF-15, the risk of cardiovascular death is comparable to those with high NPs [62]. This finding confirms the role of GDF-15 as an intermediate marker of inflammatory and multi-organ injury.

Interleukin-6 (IL6) and interleukin-1ß (ILß) are the most notorious members of the cytokines family. They are produced by activated macrophages and they are involved in several inflammatory and immunity processes [64]. They contribute directly to myocytes damage, to inflammatory burden elevation and to cardiac damage and remodeling. Moreover, cytokines impair skeletal muscle metabolism and circulation [65]. Notably, the IL-1 inhibitor ‘Anakinra’ is able to reduce hospitalization by improving exercise tolerance, treadmill parameters and quality of life in those with HFpEF [66].

Tumor necrosis factor α (TNFα) is another interleukin highly expressed in HFpEF. It correlates with adverse outcomes in a cross-sectional analysis [67]. However, several confounding factors related to the immune system may influence its levels. In stable HFpEF patients, it correlates with atrial dimension and diastolic dysfunction degree and provides additional information compared with only NPs. In the Health ABC study, it correlated with HFpEF, but it did not provide further prognostic information. Finally, anti-cytokine treatment with specific antibodies such as etanercept did not improve quality of life nor outcome [68,69].

**Endothelial dysfunction**—Microcirculation and endothelial cells are two important features for HFpEF occurrence and microvascular dysfunction is one of the most common therapeutic targets. Dysfunctional endothelium increases the expression of adhesion molecules such as vascular cell adhesion molecules (VCAM), induced cell adhesion molecules (ICAM) and E-selectin that activates von Willebrand and other prothrombotic factors. Therefore, tissue growth factors (TGFs) and insulin growth factors (IGFs) are two other items of vascular alteration and increased proliferation [70]. The prothrombotic cascade is also emphasized by several coagulation alterations involving factor V and VII, tissue plasminogen activator (TPA), inducing endothelial damage and loss of vascular integrity. Vascular, coagulative and thrombotic alterations may lead to a progressive microvascular obstruction, capillary obliteration and loss of capillary integrity [71]. These processes induce increased vascular resistance and enhanced cardiac workload at both systemic and pulmonary districts. Therefore, vascular damage is characterized by intima and media hyperplasia, disarray of smooth muscle cells and intimal fibrosis, ultimately leading to progressive capillary reduction and narrowing. All of these features reduce nitric oxide (NO) production and its mediator ‘guanosine monophosphate cyclase’ (GMPc), causing vasoconstriction, reduction in viscoelastic properties and altered oxygen consumption and utilization with increased oxidative stress [72,73]. Unfortunately, no reliable blood biomarker exists to measure these processes and only in vitro studies can document these endothelial alterations. Nevertheless, a direct GMPc activator ‘Vericiguat’ is capable of improving vascular tone and of reducing cardiac stiffness. A reliable marker of vasoconstriction and vascular tone is endothelin-1 (ET1), directly secreted by the endothelial cells in response to renin angiotensin system activation, hyperglycemia, hypertension and systemic inflammation. It is considered the most powerful vasoconstrictor factor and it is highly expressed in pre- and post-capillary and primary pulmonary hypertension, severe hypertensive status and HF irrespective of EF [74]. ET1 levels were predictive of all causes of mortality and they were associated with increased hospitalization rates in a longitudinal study of HFpEF [75]. Therefore, in RELAX analysis, ET1 correlates with reduced exercise oxygen consumption and it is significantly associated with higher NT-proBNP and galectin-3 levels [76].

Plasminogen activator inhibitor (PAI-1) is the main inhibitor of tissue plasminogen activator and the intrinsic fibrinolytic system. It is increased in patients with HFpEF in association with D-dimer levels, suggesting an association with prothrombotic and procoagulant states in this setting [77]. In the LURIC study, it is a prognostic index of mortality and CV events, although a longitudinal study confirmed only an association with markers of renal damage and NPs [78].

Insulin growth factor binding protein (IGFBP) is associated with inflammation, cell adhesion and senescence. It is increased according to left atrial dysfunction and dilatation reflecting diastolic dysfunction in HFpEF [79]. In a machine learning study, in subjects with a high inflammatory phenotype, elevated comorbidity burden and renal dysfunction, it is elevated and associated with increased hospitalization risk [80]. In the I-PRESERVE trial, IGFBP was associated with an increased risk of CV events and HF severity [81]. Finally, in asymptomatic patients with LV hypertrophy, IGFBP identifies subjects with altered diastolic function suggesting a role in early identification and screening of HFpEF [82] (Figure 2, graphical abstract).

## 4. Metabolomic Signature

Metabolic dysfunction plays an important role in systemic processes which lead to HF. There is a complex interconnection between the myocardium and peripheral tissues and organs. Relevant studies show distinctive metabolic profiles that contribute to the severity of HF [83]. However, while HFpEF is associated with indices of increased inflammation and oxidative stress, impaired lipid metabolism, increased collagen synthesis, and downregulated nitric oxide signaling, HFrEF clearly appears to be dependent on short-chain acylcarnitine oxidation, having greater FA adsorption (bile salts), transport and branched-chain amino acid catabolic deficiency [83,84].

The most studied metabolites that are involved in the metabolic profile of HFpEF are serine, lysophosphatidylcholine (LPC), kynurenine and cystine, hydroxyproline, lactate, cGMP, symmetric dimethylarginine (SDMA), arginine, cAMP and acylcarnitine [83].

In HFpEF mouse models, serine deficiency has been associated with inflammatory response and oxidative stress [85,86]. Serine is a non-essential amino acid and has different physiological functions. Serine-derived glycine is used in nucleotide synthesis. Serine is also a precursor for the synthesis of lipids, such as phosphatidylserine and sphingolipids. Thus, it is an important factor for the synthesis of nucleotides, proteins, and lipids required for cell proliferation. Serine is an allosteric activator of pyruvate kinase (PKM2). Pyruvate kinase catalyzes the last reaction of glycolysis, converting phosphoenolpyruvate (PEP) to pyruvate and producing ATP. Serine synthesis starts from the glycolytic intermediate 3-phosphoglycerate (3-PG) [87]. As serine levels increase, serine-dependent PKM2 activation could provide a feedback loop that restores the glycolytic flux in growing cells [88]. Endothelial cells use glycolysis to quickly produce energy, which helps them to adapt to changes. More metabolic intermediates are produced through glycolysis, affecting cell regulation and survival.

Through one-carbon metabolism in macrophages, serine is critical for the generation of phospholipid, biosynthesis of purine and thymidine, and production of methyl donor of S-adenosyl-methionine (SAM) and cellular glutathione. Serine is essential for the production of proinflammatory cytokines in M1 macrophages [89]. In the inflammatory process, the activation of serine proteases induces a serine deficiency. Cathepsin G is another serine protease of PMN azurophilic granules that hydrolyzes several types of proteins. Cathepsin G exerts strong pro-inflammatory effects with vascular and systemic impact [90]. Elastase is the most involved serine protease in the azurophilic granules of polymorphonuclear cells (PMN, or neutrophils). When discharged upon PMN activation elastase, it has a direct effect on the degradation of collagen, elastin and fibronectin. These processes could represent the potential basis of HFpEF development and are strictly related to the reduced myocardial compliance with diastolic dysfunction and consequent LA dilatation. Similarly, few studies found a significant increase in hydroxyproline. It is produced via hydroxylation of proline by prolyl hydroxylase and has an important role in maintaining the stability of collagen; thus, its dysregulation may contribute to the myocardial fibrosis in HFpEF [91,92].

In endothelial cells, NO is generated by endothelial nitric oxide synthase (eNOS) through the conversion of its substrate, L-arginine, to L-citrulline. Arginine is known to act as a substrate for NO production by endothelial cells [89]. Low levels of arginine reflect high concentrations of endogenous nitric oxide synthase inhibitor; SDMA, asymmetric dimethylarginine (ADMA) and N-monomethylarginine (NMMA) are associated with worsening renal function and microvascular dysfunction with reduced vasodilatory properties [93,94].

The activation of cGMP precursors via the natriuretic peptide pathway increases the cGMP levels and a subsequent protein kinase G (PKG) [83]. Furthermore, the cGMP/PKG signaling cascade phosphorylates many sarcomeric and cytosolic proteins. Downregulation of myocardial cGMP-PKG signaling in HFpEF is related to reduced myocardial brain-type NP (BNP) expression and increased microvascular inflammation and oxidative stress, which impair both the NP-cGMP and NO-cGMP axes. Decreased levels of cGMP in HFpEF, and subsequently of PKG, were associated with increased resting tension and myocyte stiffness. This feature leads to titin phosphorylation reduction that modulates the passive stiffness of cardiac muscle, thus acting as a passive diastolic distention alteration [95].

Another finding was high levels of cystine, that is, an indirect index of inflammation [96]. Cystine enters inside the cell and then it is reduced to cysteine, which is involved in the synthesis of glutathione (GSH). Glutathione peroxidase-4 uses GSH as a substrate to scavenge lipid peroxidation and reduce oxidative stress. Therefore, cysteine plays an important role in maintaining and transducing redox signals in the mitochondria [97]. Redox-dependent cysteine modification has been studied most extensively in cardiac tissue following ischemia/reperfusion injury, which deprives cardiac tissue of oxygen in the ischemic state and generates a ROS burst [98].

Kynurenine (Kyn) is a regulator of immune response, metabolized from tryptophan (Trp) during inflammatory conditions [99]. The Kyn pathway of Trp is the most active process of Trp metabolism and produces metabolites including kynurenic acid and nicotinamide adenine dinucleotide (NAD+). Very well known is the involvement of NAD+ in oxidative phosphorylation. The Kyn pathway is initiated by the enzymes’ tryptophan 2,3-dioxygenase (TDO) and indoleamine 2,3-dioxygenase (IDO and IDO2). Kyn is increased in HFpEF and has a role in the regulation of inflammation response mediated through the function as a ligand of the aryl hydrocarbon receptor (AhR) and as a transcription factor that controls local and systemic immune responses.

cAMP is produced via β-AR signaling and is inhibited via AMP-hydrolyzing enzyme phosphodiesterases, in which at least five families are expressed in the heart (PDE1, PDE2, PDE3, PDE4 and PDE8). Low cAMP levels in HFpEF suggest impaired cell signaling. Obese HFpEF patients showed an increased turnover of β-adrenergic r (β-AR) microdomains due to altered β-AR expression levels, blunted β-AR responsiveness and impaired β2-AR-coupled PDE activity [100]. Current signal cell dysregulation may alter both intracellular calcium (Ca) membrane signal and myocyte energetic process linked to glycogenolysis and lipolysis.

Additionally, HFpEF patients displayed elevated concentrations of medium- and long-chain acylcarnitines [101]. The acyl derivatives play a key role in fatty acid uptake and mitochondrial metabolism. The myocardial substrates used in ATP production in patients with preserved ejection fraction are mostly a combination of lipoprotein-derived fatty acid (LpFA) and fatty acid (FA). Biochemically, long-chain acylcarnitines are intermediates in the fatty acid ß-oxidation pathway; they are long-chain fatty acids esterified to carnitine. An increase in acylcarnitines may imply inefficient β-oxidation, which may be attributed to defects in mitochondrial FA oxidation enzymes [102,103].

Furthermore, inefficient β-oxidation leads to other metabolic pathways with an increased consumption of ketones and glutamate. The higher metabolite plasma concentrations, such as acetate and 3-hydroxybutyrate, are the markers of the worsening of heart function [104].

However, the accumulation of long-chain acylcarnitines can have a toxic effect on the phospholipids sarcolemma. These acylcarnitines interact with different ion channels and produce cardiac arrhythmias [105,106].

Low levels of lysophosphatidylcholine suggest a dysregulated phospholipid metabolism [83]. Lysophosphatidylcholine (LPC), also called lysolecithin, is a class of lipid biomolecule derived from the cleavage of phosphatidylcholine (PC) via the action of phospholipase A2 (PLA2). Phosphatidylcholine is required for the assembly of VLDLs and chylomicrons. Moreover, small alterations in phospholipid levels appear to have large implications related to the metabolic syndrome and fatty acid oxidation signaling [107]. Dysregulated lipid metabolism could drive adipose accumulation around pericardium and muscle compartments. Additionally, circulating fatty acids impair insulin sensitivity through binding to the plasma membrane toll-like receptor 4 (TLR4) in tissues of obese animals. This process results in the activation of signaling proteins, such as the inhibitor of nuclear factor-κB kinase, c-Jun N-terminal kinase and mitogen-activated protein kinase, which negatively dysregulate the metabolic axis of macrophage and favor a setting of chronic inflammation [108] (Table 2).

In order to reduce the clinical variability of metabolomic results, the optimal approach could be sharing the data within different research groups, such as Biocrates, Metabolomics Society, Consortium of METabolomics Studies (COMETS) and Phenome and Metabolome aNalysis (PhenoMeNal) [84].

## 5. Circulating MicroRNA Evidence

The non-coding genome which indicates small and long non-coding RNA is involved in gene regulation. Multi-microRNA is 21–22 nucleotide single-stranded RNAs that bind complementary messengers leading to degradation. They have been implicated in pathophysiologic processes that conduct HFpEF. Correlated with NT-proBNP, miRNAs are highly discriminatory and have improved specificity and accuracy in identifying non-acute HF [109]. The subtype stratification, mir-24-3p, has been reported to regulate apoptosis and vascularity in ischemic heart disease; mir-503-5p has been implicated in driving cardiomyocytes specification; miR-30a-5p has been shown to regulate autophagy during myocardial injury induced by Angiotensin II and miR-106a-5p promotes hypertrophy through targeting mitofusin-2, a mitochondrial primary protein in regulating cardiac function [110]. There are also pro-hypertrophic miRNAs, such as miR-208, miR-22, miR-21, miR-25, miR-34, miR-199a, miR-212/132 and miR-23 [111]. miR-3135b and miR-3908 were significantly upregulated in HFpEF and are involved in important metabolic factors for serum lipids and blood glucose levels [112]. Few circulating miRNAs may serve as markers of response to therapy: Sucharov et al. identified a set of miRNAs (miRNA 208a-3p and miRNA-591) which were differentially expressed in HF patients who responded to beta-blockers therapy [113]. Despite the increasing literature in this setting, there is no current consensus on the choice of a specific circulating miRNA serving as an HFpEF biomarker. This is due to the lack of standardized methods and different populations analyzed in multicenter trials combining laboratory data in systematic methods with homogeneous analysis.

## 6. Conclusions

Since HFpEF is a heterogeneous syndrome characterized by multiple risk factors and several associated conditions, it could be very tricky distinguishing between the main pathophysiological driver by simply considering phenotypic classification. In this framework, a detailed laboratory screening may better elucidate the underlying mechanisms responsible for HFpEF appearance and evolution. Behind new and traditional biomarkers of inflammation, cardiovascular dysfunction and fibrosis, some emerging metabolites responsible for altered cell signals, energetic substrate and excessive immune response revealed additional diagnostic properties. The challenge of future research may systematically address the real value of clinical laboratory and metabolomic combination, to effectively initiate a precision medicine methodology.

## Figures and Tables

**Figure 1 biomolecules-13-00173-f001:**
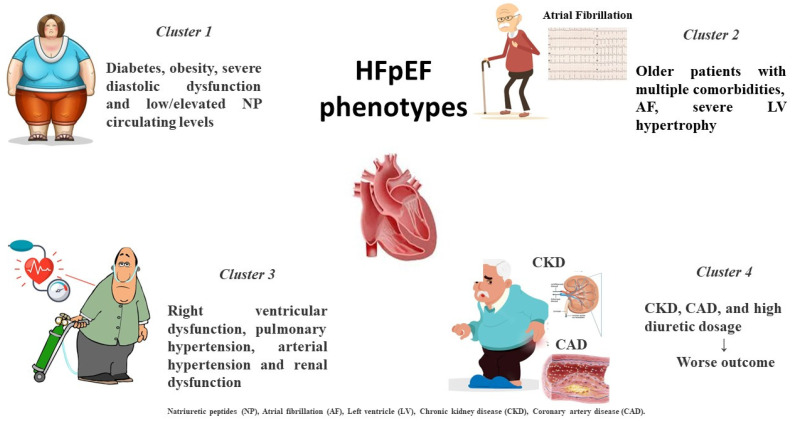
Distinct HFpEF clinical phenotypes based on clinical-presentation-associated metabolic disorders and comorbidities.

**Figure 2 biomolecules-13-00173-f002:**
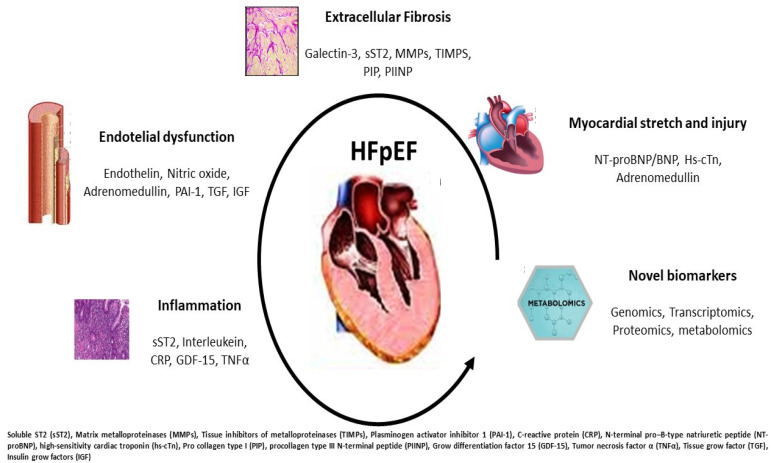
Potential pathophysiological mechanisms occurring in HFpEF: each disorder can be recognized by specific biomarker increase and overexpression. The partnership between clinical and laboratory information may better target the HFpEF profile.

**Table 1 biomolecules-13-00173-t001:** Circulating biomarkers responsible for cardiac remodeling reflecting myocardial injury, collagen overexpression, inflammation, and vascular damage.

	Name of Biomarker	Mechanism of Action
Markers of myocardial injury	↑↑ High sensitivity troponin	The final results of microvascular dysfunction, and subendocardial layer damage due to systemic oxygen reduction.
	↑ Natriuretic peptides	Related to diuresis and natriuresis which favor congestion reduction and euvolemia.
	↑↑ Adrenomedullin	Regulatory peptide produced by endothelial and smooth muscle cells with antiproliferative vasodilatatory and antiapoptotic effects.
Markers of extracellular fibrosis	↑↑ Galectin-3	Inflammatory and pro-fibrotic processes
	↑↑ Soluble ST2	Produced by myocardial cells, but smooth muscle cells and endothelium are also capable of synthesizing the peptide in relation to congestion.
	↑↑ Matrix metalloproteinases	Involved in collagen synthesis and collagen degradation.
	↑↑ Procollagen type I (PIP) and procollagen type III N-terminal peptide (PIINP)	Reflects collagen increase deposition and turnover.
Markers of inflammation	↑↑ CRP and pentraxin	Inducing complement and cytokine stimulation causing myocyte loss and endothelial dysfunction via NO production decrease.
	↑↑ Grow differentiation factor 15	Expressed in inflammatory chronic diseases, lung, kidney, and cardiovascular diseases and providing additional information on LV remodeling and function.
	↑↑ Intereleukin-6	Contributes through direct myocyte damage and indirect inflammatory burden elevation.
	↑ Tumor necrosis factor α	Correlates with atrial dimension and diastolic dysfunction degree.
Markers of endothelial dysfunction	↑↑ Vascular cell adhesion molecules (VCAM) and E selectin ↑ Endothelin 1	Activates von Willebrand and other prothrombotic factors secreted by the endothelial cells in response to renin angiotensin system activation.
	↑↑ Plasminogen activator inhibitor ↑↑ Insulin grow factor binding	In association with D-dimer levels suggesting an association with prothrombotic and procoagulant state.Left atrial dysfunction and dilatation reflecting diastolic dysfunction in HFpEF.

**Table 2 biomolecules-13-00173-t002:** Most common metabolomic pathways analyzed in HFpEF; different mechanisms suggest metabolic and energetic substrate alterations involving several cells including myocytes, macrophages, fibroblasts and endothelium.

	Biomarkers	Altered Cell Mechanism
Increased inflammation	↓ Serine ↑ Cathepsin G	Immunoregulatory actions: essential for production of proinflammatory cytokines in M1 macrophages stimulating the production of cytokines and chemokines.
	↑ Cystine	Key player in conditions of oxidative stress.
	↑ Kynurenine	Controls local and systemic immune responses.
Increased collagen synthesis and reduced myocardial compliance	↑ Hydroxyproline	Role of stability of collagen and this dysregulation contributes to myocardial fibrosis.
	↑ Elastase	Degradation of extracellular matrix components, including collagen, elastin and fibronectin.
	↓ cGMP/PKG signaling	Phosphorylation reduction associated with passive stiffness of cardiac muscle.
Endothelial dysfunction	↓ Arginine	Substrate for NO production by endothelial cells with reduced vasodilatory effects.
	↑ SDMA	Alternative methylation product of L-arginine associated with worsening renal function and microvascular dysfunction.
Energetic impairment	↓ cAMP	Is produced via β-AR signaling.
	↑ Acylcarnitine	Implies inefficient β-oxidation.
	↑ Tryptophan	Produces metabolites including kynurenic acid and nicotinamide adenine dinucleotide.
Metabolic lipid impairment	↓ Lysophosphatidylcholine	Is required for the assembly of VLDLs and chylomicrons.
	↓ cAMP	Involved in lipolysis.

## Data Availability

Not applicable.

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
