# Peer review of "Laboratory and Metabolomic Fingerprint in Heart Failure with Preserved Ejection Fraction: From Clinical Classification to Biomarker Signature"

_biomolecules, 2023, doi:10.3390/biom13010173_

Round 1
Reviewer 1 Report
This was an interesting and relevant review article about novel biomarkers and metabolic markers and micro RNA markers that can be used to potentially help risk stratify and aid in prognosis in heart failure with preserved ejection fraction. They may also help to identify further pharmaceutical targets for development of therapies directed against mechanisms such as fibrosis and inflammation. The authors did well to structure their review classifying biomarkers based on mechanism such as myocardial injury , inflammation, fibrosis and endothelial dysfunction. The clinical phenotypic classification of CHFpEF is also a useful clinical distinction.
These are the points I would like to address:
1) When the authors refer to pulmonary Hypertension, it would be useful to distinguish or specify post capillary pulmonary Hypertension or combined pre and post capillary hypertension , both seen in CHF, as opposed to primary pulmonary arterial hypertension
2) Left atrial dilatation, post capillary pulmonary hypertension or combined pre and post capillary pulmonary hypertension is seen in both congestive heart failure with reduced EF and CHFwith preserved EF. It is not limited to CHF with preserved EF
3) May also be prudent to point out that some of these biomarkers such as CRP, are not specific to congestive heart failure, and are seen in many chronic conditions.
Overall, was a thorough and well researched review. The article will need extensive grammatical and spelling editing before publication.
Author Response
thanks for your comments we hope they contributed to improbve the overall manuscript quality

Reviewer 2 Report
In all respects, the work is extremely interesting. A group of patients with in heart failure with preserved ejection fraction requires appropriate diagnostics including laboratory. The authors present advances in laboratory and metabolomic studies including HFpEF which is related to specific cells and molecular mechanisms dysregulation. The paper presents the contemporary state of knowledge. The authors have discussed in great detail the issues of pathophysiology, significance and laboratory diagnosis. The topic addressed by the authors has important clinical relevance. The work presents the contemporary state of knowledge is based on properly selected literature. The work in its current form except for minor linguistic corrections does not require significant changes and is suitable for publication.
Author Response
many thanks for reviewing our paper and precious comment you provided

Reviewer 3 Report
This interesting review entitled "Laboratory and metabolomic fingerprint in heart failure with preserved ejection fraction: from clinical classification to biomarker signature by Alberto Palazzuoli, Francesco Tramonte and Matteo Beltrami" is well-written, concise, and addresses its main point. The reviewer agrees that the systematic integration of clinical imaging and laboratory data may result in a precision medicine strategy with prognostic and therapeutic benefits.
However, there is no mention of the relevance of metabolomics in the beginning; it may be useful to the study to include a section in the introduction about the growing importance of metabolomics in HF and the metabolic differences between the HFpEF and the HFrEF. A few interesting papers on that topic have been written recently, and among them are: Ferro F, Spelat R, Valente C, Contessotto P. Understanding How Heart Metabolic Derangement Shows Differential Stage Specificity for Heart Failure with Preserved and Reduced Ejection Fraction. Biomolecules. 2022 Jul 11;12(7):969, De Jong KA, Lopaschuk GD. Complex Energy Metabolic Changes in Heart Failure With Preserved Ejection Fraction and Heart Failure With Reduced Ejection Fraction. Can J Cardiol. 2017 Jul;33(7):860-871.
I would propose expanding on lipid metabolism and exploring the differences between the subclasses. Here's an interesting paper: D. Murashige; C. Jang; M. Neinast; J.J. Edwards; A. Cowan; M.C. Hyman; J.D. Rabinowitz; D.S. Frankel; Z. Arany. Comprehensive analysis of fuel use by the failing and nonfailing human heart. 364-368 in Science 2020.
For completeness, I suggest including all of the arginine catabolites: asymmetric dimethylarginine (ADMA), symmetric dimethylarginine (SDMA), and N-monomethylarginine (NMMA).
Serine and glycolysis are intimately connected, and I propose including a part on the relationship between reduced serine, increased glycolysis, and vascular dysfunction.
Finally, a paragraph describing the clinical prospects might improve the appeal of the review.
Minor grammatical and typographical errors are found throughout the paper; thus, a complete proofread is recommended before resubmitting.
Line 33: change metabolomics with metabolic
Lines 180-183: adjust the sentence
Line 290: add protein after "Insulin grow factor binding"
Line 292-294: please reformat the sentence; the subject is missing.
Line 307: Lysophosphatidilcolyne, change in lysophosphatidylcholine
Line 361: Obese change with obese
Line:392: HFpEf change with HFpEF

Author Response
we appreciated the reviewer's comment and we answer most of issues arised. We did not expand the lenght manuscript and we did not compare Laboratory appraisal in HFpEF and HFrEF, that would require another different paper

Round 2
Reviewer 3 Report
The authors have effectively addressed the issues I rise with the initial manuscript
Author Response
Dear reviewer, we thank again for your comments. Now the paper was definetively revised by a native english professor and all language edits have been corrected. please find the updated version attached
